# Inflammatory cytokines, placental pathology, and neurological outcomes in infants born to preterm preeclamptic mothers

**Alexandra Sotiros**[1], **Dianne Thornhill**[2‡], **Miriam D. Post**[3‡], **Virginia D. Winn**[4‡], **Jennifer Armstrong**[2,5,6,7]*

**1** School of Medicine, University of Colorado Anschutz Medical Campus, Aurora, Colorado, United States of America, **2** Hemophilia and Thrombosis Center, University of Colorado Anschutz Medical Campus, Aurora, Colorado, United States of America, **3** Department of Pathology, University of Colorado Anschutz Medical Campus, Aurora, Colorado, United States of America, **4** Department of Obstetrics and Gynecology, Stanford University School of Medicine, Stanford, California, United States of America, **5** Department of Pediatrics, Section of Neurology, University of Colorado Anschutz Medical Campus, Aurora, Colorado, United States of America, **6** Department of Neurology, University of Colorado Anschutz Medical Campus, Aurora, Colorado, United States of America, **7** Department of Obstetrics and Gynecology, Division of Basic Reproductive Sciences, University of Colorado Anschutz Medical Campus, Aurora, Colorado, United States of America

's These authors contributed equally to this work.
‡ These authors also contributed equally to this work.
* jennifer.armstrong@cuanschutz.edu

**Data Availability Statement:** The data that support the findings of this study are available on request from the Director of the Colorado Multiple Institutional Review Board, john.

## Abstract

Preeclampsia is both a vascular and inflammatory disorder. Since the placenta is a conduit for fetal development, preeclampsia should be a presumed cause of adverse infant outcomes. Yet, the relationship of placental pathology, inflammation and neurological outcomes after preeclampsia are understudied. We prospectively examined a cohort of maternal-infant dyads with preeclampsia for maternal inflammatory cytokines at time of preeclampsia diagnosis and delivery, and fetal cord blood cytokines (IL-1β, IL-6, IL-8, and TNF-α). Placentas were analyzed for inflammatory and vascular pathologies. Neurodevelopmental assessment of infants utilizing the Pediatric Stroke Outcome Measure (PSOM) was conducted at 6-month corrected gestational age. Eighty-one maternal-newborn dyads were examined. Worse neurological outcomes were not associated with elevated maternal / fetal cytokines. Early preterm birth (gestational age ≤ 32 weeks) was associated with worse neurological outcomes at 6-months regardless of maternal/ fetal cytokine levels, placental pathology, or cranial ultrasound findings (OR 1.70, [1.16–2.48], p = 0.006). When correcting for gestational age, elevated IL-6 approached significance as a predictor for worse developmental outcome (OR 1.025 [0.985–1.066], p = 0.221). Pathological evidence of maternal malperfusion and worse outcomes were noted in early preterm, although our sample size was small. Our study did not demonstrate an obvious association of inflammation and placental pathology in preeclampsia and adverse neurodevelopmental outcome at 6-month corrected age but does suggest maternal malperfusion at earlier gestational age may be a risk factor for worse outcome.

heldens@cuanschutz.edu, upon reasonable
request. The data are not publicly available due to
their containing information that could
compromise the privacy of research participants.

**Funding:** NIH BIRCWH K12 KD HDO57022 (JA),
AHA SouthWest Affiliates Clinical Research Grant
10CRP3670014 (JA), and Maternal and Child
Bureau 340B program H30MC24049 (JA). The
funders had no role in study design, data collection
and analysis, decision to publish, or preparation of
the manuscript.

**Competing interests:** The authors have declared
that no competing interests exist.

## Introduction

Preterm birth is a known risk factor for long term neurological disabilities in the neonate [1, 2]. Medically indicated preterm births make up 31% of all preterm births with preeclampsia accounting for 23–43% of these indications [3]. Despite the high proportion of preterm births resulting from preeclampsia, preeclampsia itself is historically a debated risk factor for developmental delays and neurologic disabilities in the child [4–6].

The pathophysiology of preeclampsia is complex and rooted in the interplay between maternal and placental factors [7] with the key characteristics of maternal inflammation and vascular etiologies [8–10]. Proposed pathophysiologies involve the expression of maternal proinflammatory factors that manifest in the clinical expressions including hypertension, proteinuria, and transaminitis [11–14]. Many proinflammatory markers have been implicated in preeclampsia including IL-6, IL-8, IL-10, IL-17, and TNF-α [8, 15–19]. Yet, the presence of these factors alone does not characterize or indicate disease, as they also appear in other pathologies as well as normal pregnancy [20].

The maternal pathophysiology impacts fetal physiology through the filter of the placenta. Triggers of the fetal inflammatory response [21] are not completely understood, as there is contradictory evidence as to whether cytokines directly cross the placenta-blood barrier and emerge on the fetal side or if the fetal inflammatory response is indirectly triggered [22]. Regardless, the developing brain is sensitive to these immunologic signals [23] which have the potential to influence neurologic phenotypes [24, 25]. Human studies propose causal connections because of observing patterns of inflammation and outcomes, for instance that maternal IL-6 is associated with decreased working memory [26]. Animal studies seek to prove these connections, and it has been demonstrated how maternal placental inflammation (notably IL-1, IL-17a) directly lead to perinatal brain injury [27, 28]. The evidence is convincing that maternal proinflammatory states affect the developing fetal brain.

The argument for preeclampsia being an independent risk factor for poor neurological outcomes is becoming clearer. In a recent analysis by Sun et al [29] of a large prospective cohort of Norwegians (980,560 children; 28,068 exposed to preeclampsia) there was a risk associated with preeclampsia and long-term neurodevelopmental disorders in the children including attention deficit hyperactivity disorder, autism spectrum disorders, epilepsy, and intellectual disability. The pathophysiology behind this risk still needs to be investigated. There is a growing body of evidence that suggests maternal inflammation plays a role in these associations by altering neurodevelopment [22, 23], though large trials are needed to confirm this [30]. Furthermore, most inflammatory studies have been limited to term (>37-week gestational age) preeclampsia.

Though the potential of impaired fetal blood flow from abnormal vascular placentation combined with placental inflammation negatively impacting the developing brain has been theorized, outcomes beyond the neonatal period have been understudied. We sought to determine if a distinct pattern of inflammatory markers combined with placental inflammatory pathology in preterm preeclampsia was associated with long-term adverse neurologic outcomes. We hypothesized that higher levels of inflammatory cytokines in the maternal and fetal circulation, combined with placental inflammatory lesions, leads to worse neurodevelopmental outcome at 6-month corrected gestational age.

## Methods

We conducted a prospective inceptional cohort study of preterm preeclampsia at University of Colorado Hospital July 1, 2010, through June 30, 2012. Women ages 18–50 years old were identified via daily census review by a trained perinatal clinical research nurse and enrolled at

time at admission to Labor and Delivery for new diagnosis preterm preeclampsia (<37 weeks gestational age at hospitalization). Preeclampsia was identified according to University of Colorado Department of Obstetrics clinical practice guidelines: in addition to having an elevated blood pressure (>140mmHg systolic or >90mmHg diastolic) measured on 2 separate times at least 4 hours apart, there must be one of the following components: 0.30 urine protein-to-creatinine ratio, elevated liver enzymes of twice the limits of normal, thrombocytopenia (platelets <100,000 X10$^9$/L), or elevated serum creatinine >1.1 mg/dL (or a doubling of creatinine), severe persistent right upper quadrant or epigastric pain, pulmonary edema, or new-onset headache (without visual disturbances) all of which cannot be explained by another etiology. Twin gestations were rare but included to enrich the cohort. Nonviable pregnancies were excluded. The Colorado Multiple Institutional Review Board approved this study (09-1107/11-1409) in accordance with the 2004 Declaration of Helsinki, and signed informed consent was obtained from participants.

A trained perinatal research nurse collected demographic and clinical data from the University of Colorado Hospital perinatal database using a standardized survey collection form ascertained from the electronic medical record; ambiguity of maternal data was clarified by a maternal-fetal medicine specialist (V.D.W.) and for neonatal data by a neonatal neurologist (J. A.). Gestational age was ascertained by documentation of last menstrual period within the clinical chart. Data included: (1) demographics (maternal age, parity, and ethnicity; newborn sex and birthweight; prenatal care onset); (2) maternal predictors (preexisting medical conditions, gestational diabetes, previous preeclampsia); (3) route of delivery (vaginal or cesarean); and (4) neonatal complications (respiratory distress, hemodynamic failure, sepsis, hypoglycemia, seizures).

## Cytokine analysis

Maternal venous blood samples were collected at time of enrollment and within 2 hours after delivery. Fetal cord blood venous samples were obtained immediately after delivery by trained perinatal research nurses with experience in venous cord blood collection. Samples were collected in EDTA tubes and centrifuged for 20 minutes at 1600g at 4˚ C, then transferred and centrifuged again. Platelet-poor plasma was aliquoted, frozen, and sent for cytokine analysis (IL-1β, IL-6, IL-8, and TNF-α) via the Luminex multicode assay platform (Luminex Corp, Austin, TX). Elevated cytokines were defined a prior as an individual level greater than the pooled sample mean.

## Placenta analysis

Assessments were conducted by a placental pathologist (M.D.P). Placentas were immediately collected after delivery, placed in 10% neutral buffered formalin, and examined within 72 hours. A minimum of four formalin-fixed paraffin-embedded blocks were generated, including cross sections of umbilical cord, free fetal membranes, and full-thickness placental parenchyma; 5-micron sections from each block were stained with hematoxylin and eosin. Placentas were evaluated for evidence of fetal- or maternal-derived inflammation. Fetal derived inflammation, referred to as the fetal inflammatory response, was defined as the presence of inflammatory cells in the walls of fetal vessels of the chorionic plate (chorionic plate vasculitis), of the umbilical cord (vasculitis or phlebitis), or extending into Wharton's jelly (funisitis). Chronic villitis, or villitis of unknown etiology was defined as the infiltration and destruction of chorionic villi by maternal lymphocytes. Histologic chorioamnionitis was defined as inflammation composed of maternal acute inflammatory cells (polymorphonuclear cells) within the amnion or chorion. Placentas were also evaluated for evidence of maternal or fetal malperfusion.

Maternal vascular malperfusion was histologically characterized by the presence of hyperma-ture villi, decidual vasculopathy, and increased syncytial knots. Fetal vascular malperfusion, formerly termed fetal thrombotic vasculopathy, manifests as clusters of avascular villi, fibrin deposition within chorionic plate or stem villous vessel walls, and villous stromal vascular kar-yorrhexis (formerly referred to as hemorrhagic endovasculitis). Of note, placental diagnoses were rendered before adoption of the 2015 Amsterdam consensus [31].

## Postnatal evaluation

For cohort continuity, postnatal evaluations were performed per our clinical research program protocol as previously described [32]. Specifically, newborns underwent cranial ultrasounds at least 4 days after birth. Infants underwent standardized neurological examinations (Pediatric Stroke Outcome Measure [PSOM]) at 6-month corrected gestational age by a neonatal vascu-lar neurologist highly familiar and experienced with this tool (J.A.). This examiner was also blinded to cranial ultrasound, cytokine and placental pathology results. Although validated for pediatric stroke studies, the PSOM is a valuable tool for quantifying motor, speech, and cogni-tive/behavioral disability related to acquired brain injury, including that from preterm birth, that can be adapted for children <2 years old (Supplement 1). Subscale scoring (graded on level of severity) is 0 (no deficit), 0.5 (mild deficit, normal function), 1 (moderate deficit, decreased function), or 2 (severe deficit, decreased function); the total PSOM score ranges from 0 (no deficit) to 10 (maximum deficit) [33].

## Statistical analysis

Paired comparisons for dichotomous variables were made using Chi-Square analysis (or Fish-er's exact test for small sample sizes) and for continuous variables using Mann-Whitney U and Wilcoxon Signed Rank tests due to the non-normal distribution of values and the unequal var-iances between groups. Pearson correlations were used to examine the relationship between gestational age, PSOM outcome and cytokine levels. Gestational age was analyzed as both a continuous variable as well as dichotomized into defined clinical classifications for paired comparison: "early preterm" (< 32 weeks) and "moderate/late preterm (32–36 6/7 weeks) [34, 35]. To assess predictors of PSOM outcomes (no or mild deficit vs. moderate to severe deficit), univariate logistic regression was used to calculate odds ratios (ORs) and 95% confidence intervals (CIs). Univariate predictors included demographic, clinical as described in Table 1, as well as cytokine and placental predictors described above. Multivariate modeling was intended with predictors $p < 0.01$ included in the final model but could not be executed due to lack of adequate number of significant univariate predictors (see Results). Finally, based on "male disadvantage" effect seen with preterm birth cohorts and adverse neurodevelopmental outcomes [36–39], we added descriptive reports for the male infants. Small sample size and risk of Type II error inflation prevented any statistical analysis of male sex beyond the descrip-tives. IBM SPSS Version 27 was used for all statistical analysis. All tests were two-sided with $p < 0.05$ considered significant. Missing data were coded as null values; in the case of missing cytokine values, undetectable or absent cytokine levels were assigned zero value for effect max-imization in the overall analysis.

## Results and discussion

Eighty-one maternal-newborn dyads were examined (Table 1). All mothers delivered within 48 hours of presentation. As standard care at our institution, the vast majority (N = 70; 92%) of women received magnesium for fetal neuroprotection before delivery. The majority of women were Caucasian (N = 67; 88%), with approximately 40% of those with Hispanic

**Table 1. Demographic and clinical characteristics of preterm preeclamptic participants.**

| Characteristic | N = 76 preeclamptic mothers |
|---|---|
| Maternal age (mean, range) | 28 years (18–45) |
| Nulliparity [N = 75] | 50 (67%) |
| Maternal race | |
| Caucasian | 67 (88%) |
| Hispanic ethnicity | 26 (34%) |
| Black | 7 (9%) |
| Other/Unknown | 2 (3%) |
| Gestational age at birth Age (mean, range) | 33.3 weeks (24.8–39.4) |
| Magnesium | 70 (92%) |
| Diabetes | 12 (16%) |
| Gestational diabetes | 4 |
| Type 1 diabetes | 4 |
| Type 2 diabetes | 4 |
| Chronic Hypertension | 4 (5%) |
| Smoker at time became pregnant | 10 (13%) |
| Cesarean section | 41 (54%) |
| Male newborn | 45 (56%) |
| Twin gestation | 5 (7%) |

ethnicity (N = 26/67). Reflective of the overall population of the Rocky Mountain region, there were no Asian women or non-Hispanic ethnicities enrolled. There was a slight predominance of male infants (N = 45; 56%), although this was not statistically different. Twelve early preterm neonates (41%) were male sex while 33 moderate/late preterm neonates (66%) were male; this distribution was not statistically significant compared to female newborns. Cranial ultrasounds, although ordered, were not performed on all newborns (N = 47; 58%). Of the eighty-one neonates entered in the study, only 31 (38%) presented to their research appointment at 6-month corrected age.

Of the 47 newborns with cranial ultrasounds, only 4 had abnormal results reported: Intraventricular Hemorrhage Grade I (N = 2), Grade II (N = 1) and of unknown grade (N = 1). None of those infants scored moderate-severe on PSOM at their 6-month appointment. However, three neonates with "normal" cranial ultrasounds scored moderate-severe at their 6-month appointment.

Of those 31 infants with 6-month PSOM score available, one-third (N = 10; 32%) had adverse neurodevelopmental outcomes with PSOM score of 1 or higher. Gestational age was positively correlated with improved neurological outcome (r = -0.39; p = 0.03), with marked inflection point of outcomes differences seen at 32-week gestational age (Fig 1).

Maternal cytokines at enrollment and delivery, and fetal cord blood cytokines at delivery, are presented in Table 2. Maternal IL-6 and IL-8 increased overall from enrollment to delivery. When analyzed as continuous variable, increased maternal IL-8 at enrollment was correlated with increased gestational age (Table 3). Similarly, increased maternal IL-6 and IL-8 at delivery were correlated with increased gestational age. When gestational age was analyzed as a dichotomous variable, maternal IL-6 and IL-8 at enrollment were significantly higher (p = 0.01 and p = 0.02, respectively) in moderate/late preterm (Table 4). Maternal IL-8 at delivery was also significantly higher in the moderate/late preterm group (p = 0.009). Although no correlation was seen with continuous gestational age (all p>0.15), fetal cord blood IL-6 and IL-8 were significantly higher in moderate/late preterm (p = 0.009 and p = 0.003, respectively).

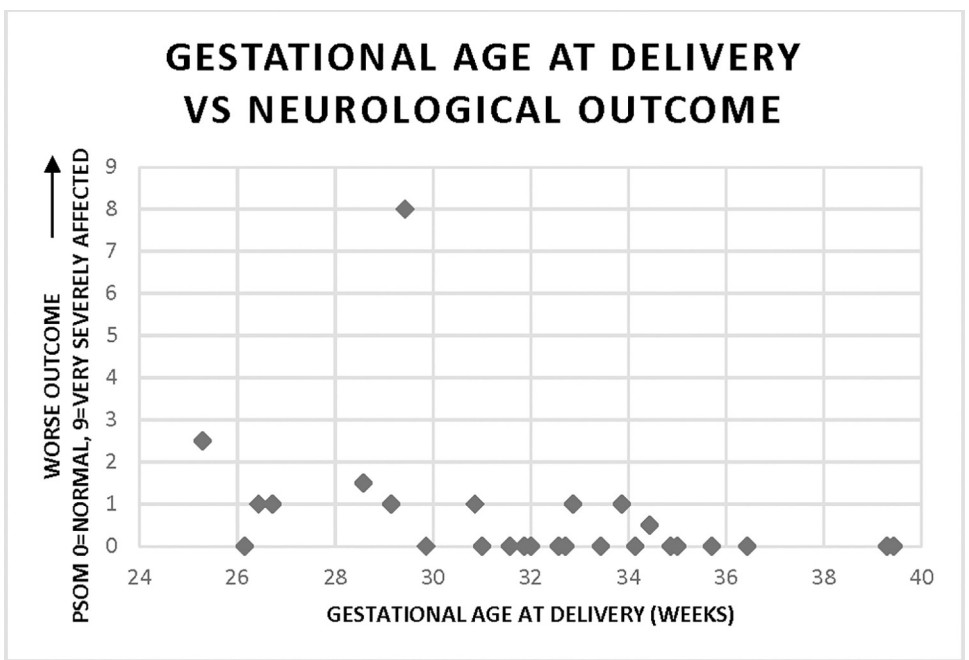

**Fig 1. Correlation of neurological outcome at 6 months with gestational age at delivery (r = -0.39, p = 0.03).** Higher score denotes worse outcome (0 = no disability, 0.5 = mild disability, 1.0 = moderate disability, ≥ 1.5 = severe disability.

Sixty-four placentas had pathological analysis available. Chronic villitis was seen in 5 cases (8%). Fetal inflammatory response was seen in 3 cases (5%); all these neonates were male. In two of these cases with fetal inflammatory response, there was also histologic chorioamnionitis present. There were no cases in which there was both fetal inflammatory response and chronic villitis. Of note, of the 22 placentas that were early preterm, only one showed evidence of

**Table 2. Mean and median values of maternal cytokines at enrollment and delivery and fetal cord blood at delivery.**

| Cytokine (pg/ml) | Mean | Median |
|---|---|---|
| IL-1β | | |
| Maternal enrollment | 0.85 (SD 0.63, range 0.00–4.00) | 0.70 |
| Maternal delivery | 0.95 (SD 0.69, range 0.00–3.60) | 0.70 |
| Fetal cord blood | 1.41 (SD 3.90, range 0.00–26.00) | 0.70 |
| IL-6 | | |
| Maternal enrollment | 9.95 (SD 16.56, range 1.5–110.30) | 5.35 |
| Maternal delivery | 39.16 (SD 38.28, range 3.00–212.80) | 26.40 |
| Fetal cord blood | 33.31 (SD 95.25, range 0.00–607.30) | 11.70 |
| IL-8 | | |
| Maternal enrollment | 5.24 (SD 3.78, range 0.00–17.00) | 4.25 |
| Maternal delivery | 9.22 (SD 5.98, range 1.56–29.80) | 8.35 |
| Fetal cord blood | 56.46 (SD 194.54, range 0.00–1290.60) | 19.30 |
| TNF-α | | |
| Maternal enrollment | 8.63 (SD 5.36, range 2.00–36.70) | 8.20 |
| Maternal delivery | 9.77 (SD 6.96, range 2.00–46.20) | 8.20 |
| Fetal cord blood | 14.26 (SD 4.41, range 0.00–28.90) | 13.80 |

**Table 3. Correlation of gestational age as a continuous variable vs cytokine level (P < 0.05 significant).**

| Cytokine (pg/dl) | Maternal Cytokine at Enrollment vs Gestational Age | | Maternal Cytokine at Delivery vs Gestational Age | | Fetal Cord Blood Cytokine vs Gestational Age | |
|---|---|---|---|---|---|---|
| | Pearson Correlation | P-Value | Pearson Correlation | P-Value | Pearson Correlation | P-Value |
| IL-1β | -0.11 | 0.42 | -0.18 | 0.22 | 0.13 | 0.41 |
| IL-6 | 0.09 | 0.51 | 0.34 | 0.02 | 0.22 | 0.16 |
| IL-8 | 0.342 | 0.01 | 0.35 | 0.01 | -0.10 | 0.54 |
| TNF-α | 0.23 | 0.09 | 0.27 | 0.06 | 0.18 | 0.26 |

chronic villitis and none showed evidence of fetal inflammatory response. Maternal vascular malperfusion was seen in 22 placentas, and were much more frequent in the early preterm group (p = 0.004). Four subjects (6%) with placental maternal malperfusion had moderate-severe PSOM score, which was not significantly different from those without maternal malperfusion (p = 0.65). There was evidence of fetal malperfusion in only three placentas (4%); none of these were associated with a moderate-severe PSOM score. Seven infants (11%) without evidence of placental inflammation scored moderate-severe at 6-month PSOM.

Given the small numbers, statistical analysis between cytokine elevation and placental pathologies could not be performed, although patterns did emerge. For instance, when chronic villitis was present (N = 5), higher maternal levels of IL-6 (N = 3; 60%) were usually present rather than in fetal cord blood. Although only 3 cases of histologic fetal inflammatory response were seen, both IL-6 and IL-8 were elevated in the fetal cord blood (N = 2; 67%), but not elevated in maternal samples. Interestingly, when there were no histological signs of inflammation (N = 56), higher levels of maternal IL-6 at enrollment (N = 17; 30%), delivery (N = 16; 29%), and fetal cord blood (N = 13; 23%) were sometimes seen; similarly, higher levels of maternal IL-8 at enrollment (N = 19; 54%), delivery (N = 18; 32%) and fetal cord blood (N = 16; 29%) were also seen. Unlike IL-6 and IL-8, there were no cases of increased IL-1β or TNF-α and placental pathologies, nor any cases of increased IL-1β or TNF-α when no placental inflammation was present.

**Table 4. Mean cytokine levels at early preterm (< 32 weeks gestational age) vs moderate/late preterm (≥ 32 weeks gestational age); P < 0.05 significant.**

| Cytokine (pg/dl) | Early Preterm | Moderate/Late Preterm | Z | P-Value |
|---|---|---|---|---|
| IL-1β | | | | |
| Maternal enrollment | 0.90 | 0.81 | 0.87 | 0.38 |
| Maternal delivery | 1.01 | 0.91 | 0.64 | 0.52 |
| Fetal cord blood | 0.84 | 1.58 | 0.89 | 0.38 |
| IL-6 | | | | |
| Maternal enrollment | 9.60 | 10.45 | -2.49 | 0.01 |
| Maternal delivery | 26.94 | 46.03 | -1.83 | 0.07 |
| Fetal cord blood | 7.57 | 41.12 | -2.62 | 0.009 |
| IL-8 | | | | |
| Maternal enrollment | 4.13 | 6.15 | -2.41 | 0.02 |
| Maternal delivery | 6.96 | 10.49 | -2.61 | 0.009 |
| Fetal cord blood | 43.73 | 60.31 | 2.96 | 0.003 |
| TNF-α | | | | |
| Maternal enrollment | 7.52 | 9.62 | -1.45 | 0.15 |
| Maternal delivery | 7.55 | 9.15 | -1.70 | 0.09 |
| Fetal cord blood | 13.19 | 14.62 | -1.37 | 0.17 |

On univariate analysis, only early preterm was a predictor of worse neurological outcomes (OR 1.70, [1.16–2.48], p = 0.006). Maternal factors of age, parity, race/ethnicity, diabetes, chronic hypertension or smoker were not significant predictors. In addition, antenatal administration of magnesium nor male sex were not predictors of neurological outcome. Since gestational age was the only significant predictor of worse neurological outcome on univariate analysis, we were unable to perform multivariate analysis. There were no cytokine or placental predictors of worse neurological outcome within our cohort when adjusted for gestational age at birth.

## Discussion

In this prospective cohort study of maternal-infant dyads from pregnancies complicated by preeclampsia, improved neurological outcome at 6-months corrected gestational was seen with increased gestational age at birth. There appeared to be an inflection point of better neurological outcomes at 32-weeks, consistent with the definition of moderate to late preterm birth and similar to other studies [32, 34, 40]. However, elevated maternal and fetal cord blood IL-6 and IL-8 was seen with increasing gestational age at birth. Furthermore, the moderate/late preterm group in particular were significantly elevated compared to the early preterm group. Additionally, there were no associations between placental pathology and worse neurological outcomes, though early preterm was associated with histologic evidence of maternal malperfusion. Taken all together, our findings do not support our–and others–hypothesis that inflammation manifesting as increased serum cytokines and placental pathology are risk factors for worse neurological outcomes in pregnancies complicated by preeclampsia. This discrepancy may stem not only from our small numbers, from the observation that there are many etiologies and forms of preeclampsia [41] which make it challenging to correlate risks. In other words, just as there is a spectrum of disease in preeclampsia, there is likely a spectrum of risk and outcome.

Our study, though trending toward a link between maternal malperfusion and worse neurodevelopmental outcome at 6-months corrected gestational age, did not reach statistical significance. Other studies have examined preeclamptic placental pathologies in relation to neonatal outcomes, which suggests our small sample size may be hindering our results. Weiner et al found an association between maternal placental malperfusion and adverse neonatal outcomes including cerebral morbidity but did not include an analysis of inflammatory lesions in the 70 studied placentas nor examine outcomes beyond the neonatal period [42]. In a study of 76 placentas describing decidual vasculopathy patterns, smaller birthweight was correlated with more extensive decidual vasculopathy but again analysis did not include neurological morbidity [43]. In a cohort of 544 placentas, there was no correlation between maternal parenchymal inflammation (chronic villitis) and neonatal morbidity including intraventricular hemorrhage and cystic periventricular leukomalacia [44]; however, this paper did not capture other characteristics of maternal inflammatory response such as histologic chorioamnionitis. These studies, albeit important to the field, did not examine the full scope of inflammatory lesions in relation to neurological neonatal and infant outcomes as we did. Larger studies are needed to examine the trend noted here and by Vinnars et al [44], that inflammatory lesions in preeclamptic placentas do not impact neurologic outcomes.

Although not statistically significant, interesting patterns of placental pathology emerged. For instance, similar to our previous findings in preterm premature rupture of membranes [45, 46], we found no cases with both chronic villitis and fetal inflammatory response. Additionally, similar to our preterm premature rupture of membranes cohort, our two cases of histological chorioamnionitis with associated fetal inflammatory response. These data suggest

that fetal inflammation as a progression of histologic chorioamnionitis, although *in situ* fetal inflammatory response cannot be ruled out

Cytokines are complex as they correlate with harmful and protective factors. The results of our study differ from the larger observational ELGAN studies. In the analysis by O'Shea et al, neonates with elevated IL-6, IL-8, and TNF-α on two separate dates had higher odds (OR 2.4; 95% CI, 1.4–4.0, OR 2.2; 95% CI 1.3–3.8, and OR 1.9; 95% CI 1.1–3.2 respectively) of having impaired cognitive function at 2 years corrected age [47]. Differences may stem from our highly specialized population, our small sample size and low follow-up rate impairing power, and because our analysis solely examines cord blood samples.

Our prior studies and others have suggested that >32 weeks gestational age may be an inflection point for improved neurodevelopmental outcomes [35, 40, 45, 46]. Our study found the same results, with having better neurodevelopmental outcomes if born moderate/late preterm. It has been postulated that less inflammation and/or healthier placental interface may contribute to better newborn outcomes in moderate/late preterm births; our findings were not consistent with these postulates. In fact, we saw higher IL-6 and IL-8 moderate-late preterm births despite having better neurodevelopmental outcome at 6-months. Furthermore, despite maternal vascular malperfusion exclusively seen in early term placentas, there was no clear association with adverse neurodevelopmental outcomes. These findings highlight the nuances of the preterm preeclamptic inflammatory and placental environment on the preterm brain are still not understood.

Sexual dimorphism has been observed in preeclamptic pregnancies [48] but the clinical impact remains unclear. We did not see a "male disadvantage" within our cohort. Males did not have worse neurologic outcome at 6-months corrected gestational age compared to females. Additionally, male newborns did not exhibit elevated cytokine levels, nor was there higher numbers of placental pathological lesions seen in males, compared to females. This is different than in our preterm premature rupture of membranes cohort, where males had worse neurological outcomes, yet it was females that exhibited higher cytokine levels suggestive of a protective cytokine effect [32]. We suspect that in both studies small numbers contribute to the conflicting and/or insignificant results. Again, further large-scale studies of neurodevelopmental outcome, inflammation and placental pathology in distinct, understudied preterm populations such as preeclampsia and preterm premature rupture of membranes is needed.

We saw that cranial ultrasound results did not reliably predict neurological function at 6-month corrected gestational age. This is similar to our preterm premature rupture of membranes cohort [46, 49] and suggests that clinically relevant developmental findings may be affected by injury at a molecular level. Many investigators have shown that preterm MRI, including volumetric studies, are more sensitive and predictive of neurological outcomes [50, 51]. We agree with this recommendation to utilized brain MRI when available and caution against the use of routine cranial ultrasounds for prognosis aside from gross hydrocephalus or significant intracranial hemorrhage. That being said, routine brain MRI may not be feasible or practical at many institutions, especially if there is no access to MRI scanners or inadequate nursing staff. Even in our large academic Level III nursery–with adequate funding in our grant to pay for transport and execution of research MRIs–MRIs could not be obtained. This was blocked before the start date of our research protocol by neonatal clinical team and hospital administration as there was decreased comfort level to have these preterm newborns off the neonatal floor, as well as limited availability of MRI scanner time for the entire hospital. Therefore, the research MRI portion was ultimately excluded from our study. Even in the best-case research scenario with on-site perinatal research nurses and standing research ultrasound order, only about half of our infants received their cranial ultrasounds. Post-hoc interviews

with NICU staff and families uncovered that 1.) ultrasound technicians were not allowed to do bedside ultrasound as clinical team did not feel it was necessary, 2.) neurology was not felt necessary to be part of clinical decision making in a consultant or outpatient realm or 3.) late preterm infants were discharged before 4 days old and did not show to their outpatient research ultrasound. Our study highlights major institutional limitations, the importance of a valued relationship between the clinical care team and research team, as well as the importance of on-site neurological care in partnership with neonatal and pediatric care of these at-risk infants.

Our study has several limitations, the most obvious small sample size. With a larger sample, a true association between cord IL-6 and elevated PSOM may be revealed as well as answer questions about sex differences and placental pathology. Measuring PSOM at 6-month corrected gestational age also limits data on outcomes that are readily apparent but may be more apparent later in life. Additionally, there may be bias in those children who were lost to follow-up, although it is difficult to comment if those children were less affected, there were potential barriers to follow-up (such as elements related to underprivilege) [52], or a combination of all. We lacked a control group, and instead compared neurological outcomes within only the preeclamptic population limiting the understanding of background placental pathology and cytokine variability. In essence, we utilized the preeclampsia population at their own control group, with a case defined as adverse infant neurological outcome. There is no true "healthy" control group in preterm infants. However, without a pure matched control group, we do not know the absolute prevalence of placental and inflammatory measures for gestational age. In addition, gestational age was ascertained by clinical chart review, but given case mix of prenatal care could not be confirmed by dating from first trimester ultrasound. Finally, lack of adequate neuroimaging to correlate with outcomes limited the overall prognostic and associative capabilities of our study.

## Conclusions

Our study supports the distinction that 32-weeks gestational age ('moderate preterm) is a turning point for improved neurological outcomes. We found that cranial ultrasounds were unhelpful in predicting risk in this preeclamptic population. Our data suggest that maternal placental malperfusion at earlier gestational age may be an underlying factor associated with poor neurological outcomes, although more robust studies are warranted. Neither elevated cytokines nor "male disadvantage" was associated with adverse neurological outcomes. Our study was limited by sample size. Larger long-term longitudinal studies of preeclampsia are needed to adequately examine the inflammatory and vascular patterns associated with long-term neurologic outcome.

## Supporting information

**S1 File. Pediatric stroke outcome measure.**
(PDF)

**S2 File. STROBE checklist.**
(PDF)

## Author Contributions

**Conceptualization:** Miriam D. Post, Virginia D. Winn, Jennifer Armstrong.

**Data curation:** Alexandra Sotiros, Dianne Thornhill, Miriam D. Post, Virginia D. Winn, Jennifer Armstrong.

**Formal analysis:** Alexandra Sotiros, Dianne Thornhill, Miriam D. Post, Jennifer Armstrong.

**Funding acquisition:** Jennifer Armstrong.

**Investigation:** Miriam D. Post, Virginia D. Winn, Jennifer Armstrong.

**Methodology:** Dianne Thornhill, Miriam D. Post, Virginia D. Winn, Jennifer Armstrong.

**Project administration:** Jennifer Armstrong.

**Resources:** Jennifer Armstrong.

**Supervision:** Virginia D. Winn, Jennifer Armstrong.

**Validation:** Dianne Thornhill, Miriam D. Post, Virginia D. Winn.

**Writing – original draft:** Alexandra Sotiros, Dianne Thornhill, Miriam D. Post, Virginia D. Winn, Jennifer Armstrong.

**Writing – review & editing:** Alexandra Sotiros, Dianne Thornhill, Miriam D. Post, Virginia D. Winn, Jennifer Armstrong.

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
