## [Decision Letter · Decision Letter 0]

22 Jun 2021

PONE-D-21-14686

Biomarkers of inflammation, placental pathology, and neurological outcomes in infants born to preeclamptic mothers

PLOS ONE

Dear Dr. Armstrong,

Thank you for submitting your manuscript to PLOS ONE. After careful consideration, we feel that it has merit but does not fully meet PLOS ONE’s publication criteria as it currently stands. Therefore, we invite you to submit a revised version of the manuscript that addresses the points raised during the review process.

If you can address the specific concerns detailed below, then we would be willing to reconsider a revised version. Before seeing and evaluating such changes, however, we cannot guarantee that your revised article would be accepted for publication. Should you decide to revise the manuscript for further consideration here, your revisions should address all points indicated by reviewers.

Please submit your revised manuscript within 30 days. If you will need more time than this to complete your revisions, please reply to this message or contact the journal office at plosone@plos.org. Please include the following items when submitting your revised manuscript:

We look forward to receiving your revised manuscript.

Kind regards,

Claudio Romero Farias Marinho, Ph.D.

Academic Editor

PLOS ONE

3. Please amend the manuscript submission data (via Edit Submission) to include authors : “Alexandra Sotiros MS , Dianne Thornhill PhD , Miriam D Post MD and Virginia D Winn MD “

4. Please amend your list of authors on the manuscript to ensure that each author is linked to an affiliation. Authors’ affiliations should reflect the institution where the work was done (if authors moved subsequently, you can also list the new affiliation stating “current affiliation:

   a. University of Colorado Anschutz Medical Campus (CU-Anschutz),

   b. Hemophilia and Thrombosis Center,

   c. Department of Pathology

   d. Stanford University School of Medicine, Department of OB/GYN,

   e. Departments of Pediatrics (Section of Neurology), Neurology, and OB/GYN (Basic Reproductive Sciences)

   f. Children’s Hospital Colorado.” as necessary).

5. Please ensure that you refer to Figure 2 in your text as, if accepted, production will need this reference to link the reader to the figure.

Reviewers' comments:

Reviewer's Responses to Questions

**Comments to the Author**

1. Is the manuscript technically sound, and do the data support the conclusions?

Reviewer #1: Partly

Reviewer #2: Yes

2. Has the statistical analysis been performed appropriately and rigorously? 

Reviewer #1: Yes

Reviewer #2: Yes

3. Have the authors made all data underlying the findings in their manuscript fully available?

Reviewer #1: Yes

Reviewer #2: No

4. Is the manuscript presented in an intelligible fashion and written in standard English?

Reviewer #1: Yes

Reviewer #2: Yes

5. Review Comments to the Author

Reviewer #1: Thank you for the opportunity to read this article. The study provides interesting information about the association of inflammatory cytokines, placental injury, and neurological outcomes in newborns of pre-eclamptic pregnant women. Objectives of this study are well stated. Many important details of the methods, especially of sampling from the participants, are omitted. I expect that previous publications provide this information, but readers of this article should not have to seek out those papers in order to understand the study. Thus, it needs several major and minor changes before it can be published.

General comments:

- There are some sections that need English language editing - it would be helpful if the manuscript were re-reviewed with attention.

- I suggest changing the title, because the use of the word “biomarker” refers to the idea that the authors tested the hypothesis of these cytokines to predict some adverse event, which is not true. For example: “Inflammatory cytokines, placental pathology, and neurological outcomes in infants born to preeclamptic mothers.”

- Consider revising all abbreviations in the text, as some are mentioned without a description. The excessive use of abbreviations makes it difficult to read and understand the entire text. Authors should use abbreviations only when necessary. Following the guidelines of the journal: 1- Do not use non-standard abbreviations unless they appear at least three times in the text; 2- Keep abbreviations to a minimum.

- I strongly suggest that the authors write the article following the guidelines of STROBE, attaching the checklist as supplementary material.

Methods

- It would be helpful to know (briefly) how the women in this study were recruited - inclusion/exclusion criteria must be well defined; when were pregnant women recruited - at delivery?

- Line 77-78: The authors stated that pre-eclampsia was identified according to standard practice. What standard practice? WHO? Missing reference.

- Line 84-85: Could the authors comment on why multiple pregnancies were included in the analyzes? Wouldn't the chances of these fetuses develop neurological problems be greater, causing a bias in the analysis?

- Line 104: Were the professionals blinded to the placental analyzes?

- It is very important to note that several sections of the study, but especially “Cytokine Analysis” and “Postnatal evaluation” are written identically to the previous study of the group (ref. 31), without even mentioning that this had already been done before. I don't believe this is correct! I suggest that the authors cite previous studies.

- How was gestational age ascertained? This is important because the authors use the gestational age of 32 weeks for cut-off.

- Line 134: The “Statistical Analysis” section should be better written - data on the adopted confidence interval, p-value, are missing. I suggest that the authors read the journal's guidelines: section “Reporting of statistical methods”.

- Line 137-139: The authors stated that: “Gestational age at delivery was dichotomized into ≤32 weeks and >32 weeks for paired comparisons to maintain comparative consistency with our previous study cohorts (30, 31)”. However, this does not seem to me to be a plausible justification. A more biological/clinical justification is needed. Could the authors comment?

Results

- Line 153-159: The authors present the results of all the cytokines evaluated, but Table 2 shows only two (IL-6 and IL-8). I suggest adding the entire result in the table, together with the p-value and the description in the legend of the statistic used.

- In addition to the previous comment, tables 1, 2 and 3 need adjustments. The tables must have appropriate titles and legends, with a description of any abbreviation, and the statistics used; and the p-value must be indicated in the table. Authors should read the journal's guidelines: 1- Place each table in your manuscript file directly after the paragraph in which it is first cited (read order); 2- Tables require a label (e.g., “Table 1”) and brief descriptive title to be placed above the table. Place legends, footnotes, and other text below the table.

- Line 160-164: I did not understand why the authors describe these results only for males. Could you explain?

- Line 181-184: The way these results are presented seems to me confused. I suggest that the authors present a table or figure that contains these results, with a complete description of the statistical method used.

- Line 187: Figure 1 or 2?

Discussion

- Line 227-229: Only corroborating with a previous comment, the authors presented results only for male newborns and in this paragraph of the discussion they describe only about female newborns. If the authors consider gender to be important, both must have results and be discussed.

- The authors presented results on parity, diseases such as chronic hypertension and diabetes, type of delivery (Table 1) and did not discuss anything about it. Could you comment since the N of these variables is considerable?

- The authors divided their population according to their gestational age (greater or less than 32 weeks) and did not discuss anything about it. This is a key point of the study that must be discussed.

Reviewer #2: The manuscript “Biomarkers of inflammation, placental pathology, and neurological outcomes in infants born to preeclamptic mothers” by Armstrong et al, analyzed serum inflammatory markers of preeclampsia and aimed to correlate these markers and placental inflammation to neonate brain injury. The manuscript presents findings that corroborate the presence of placental inflammation and brain injury in newborns, which was correlated to gestational age less than 32 weeks. However, this reviewer believes some points need to be better explained and added to improve the paper.

# Major limitations

Material and methods

In general, the methodology was appropriated to collect the data and perform the intended analysis. According to this reviewer, the topic of the postnatal evaluation is not clear. Authors should provide more details and make clearer the information about the parameters used to determine the scores to classify the brain injury.

Results

This reviewer recommends that the following points are addressed:

1. Line 160 provides information on the percentage of male neonates with GAD < 32 weeks but there is no explanation/discussion about the relation of these observations with the pathology. Please add a statement on the manuscript to clarify the relevance of the above mentioned data.

2. In the paragraph between the lines 166-171: It was not clear for this reviewer if the analysis was performed with the placentas from GAD > 32 weeks or if both data from GAD <32 weeks and GAD > 32 weeks were considered in this analysis. Please clarify this information.

3. Line 187: Figure 1 was cited following the sentence “Within that sample at the 6-month visit (Figure 1)”. This reviewer does not understand the relationship between the neurological outcome and Figure 1, if there is any. Figure 1A represents chronic villitis and figure 1B is a figure of an umbilical artery. Please clarify the intended meaning.

4. Figure 1: As highlighted by the authors, matching healthy controls in a limitation of this study. Nevertheless, this reviewer believes that evidence of normal placental structure and umbilical artery should also be provided.

5. Regarding clinical and experimental database. This reviewer believes that in the interest of data availability, authors should make all data underlying the described finding available in the form of supplementary material.

Specific comments for figures and tables

Table 1. Magnesium levels are represented in table 1but are not mentioned in the text. Please explain the importance to measure magnesium for this analysis.

Table 2. The table displayed information regarding the IL-6, IL-8, but no information about TNF-alpha levels. This reviewer suggests including this information.

Figure 1. Chronic villitis (A) and fetal inflammatory response (B). Authors should have used arrows signalizing the observed evidence observed. Please provide a new version of Figure 1.

Figure 2 was not mentioned in the text. Authors should explain its results and discuss its relevance in the text.

Discussion

In the study, the results showed an association between gestational age at delivery ≤ 32 weeks and worse neurological outcomes at 6-month CGA. There is no association between maternal or fetal cytokine activity and neurological outcomes, even in the samples with evidence of maternal malperfusion in the placenta. The authors discuss the current methodology (cranial ultrasound) used to predict neurological function, suggesting that the injury could be at a molecular level, being the ultrasound not appropriated to predict neurological dysfunctions. This reviewer agrees, but alternative methods should be provided by the authors to fulfill this shortcome. For example, but not limited to, is the study of Setänen et al., 2016 (doi.org/10.1111/dmcn.13030) used volumetric neonatal magnetic resonance imaging to predict the neuromotor outcome in infants born preterm.

# Minor limitations

Line 167: Chronic villitis in Figure 1A

6. PLOS authors have the option to publish the peer review history of their article (what does this mean?). If published, this will include your full peer review and any attached files.

Reviewer #1: No

Reviewer #2: **Yes: **Livia Rosa-Fernandes

---

## [Author Response · Author response to Decision Letter 0]

16 Sep 2021

Please be aware, per Reviewer 1 suggestion, the Title has been changed to "Inflammatory cytokines, placental pathology, and neurological outcomes in infants born to preterm preeclamptic mothers"

Please see detailed Response to Reviewers line by line rebuttal included in uploaded documents as this was a major revision. The STROBE checklist is also included there. Thank you.

---

## [Decision Letter · Decision Letter 1]

1 Nov 2021

PONE-D-21-14686R1Inflammatory cytokines, placental pathology, and neurological outcomes in infants born to preterm preeclamptic mothersPLOS ONE

Dear Dr. Armstrong,

Thank you for submitting your manuscript to PLOS ONE. After careful consideration, we feel that it has merit but does not fully meet PLOS ONE’s publication criteria as it currently stands. Therefore, we invite you to submit a revised version of the manuscript that addresses the points raised during the review process.

Your changes to the manuscript has been re-reviewed by the original reviewers.  Reviewer No. 1 has made some suggestions that should strengthen the manuscript.  Please respond to Reviewer No. 1's suggestions.

We look forward to receiving your revised manuscript.

Kind regards,

Khursheed Iqbal, Ph.D

Academic Editor

PLOS ONE

Journal Requirements:

Additional Editor Comments (if provided):

Reviewers' comments:

Reviewer's Responses to Questions

**Comments to the Author**

1. If the authors have adequately addressed your comments raised in a previous round of review and you feel that this manuscript is now acceptable for publication, you may indicate that here to bypass the “Comments to the Author” section, enter your conflict of interest statement in the “Confidential to Editor” section, and submit your "Accept" recommendation.

Reviewer #1: All comments have been addressed

Reviewer #2: All comments have been addressed

2. Is the manuscript technically sound, and do the data support the conclusions?

Reviewer #1: Yes

Reviewer #2: Yes

3. Has the statistical analysis been performed appropriately and rigorously? 

Reviewer #1: Yes

Reviewer #2: Yes

4. Have the authors made all data underlying the findings in their manuscript fully available?

Reviewer #1: No

Reviewer #2: No

5. Is the manuscript presented in an intelligible fashion and written in standard English?

Reviewer #1: Yes

Reviewer #2: Yes

6. Review Comments to the Author

Reviewer #1: Armstrong et al did a good job of reviewing the article. The authors have an interesting data set with potential for an interesting manuscript. However, some minor revisions still need to be resolved before publication.

INTRODUCTION

• Line 54-60: The word preeclampsia is used six times in this paragraph. Consider using synonyms.

• Review the citations made in the introduction. A few sentences have no citation.

METHODS

• Line 105-106: The accuracy of gestational age (GA) is important. So, ultrasound in the first trimester of gestation can represent a better tool to access GA. Did the authors have that information? This can be added as a limitation of the study.

• Line 141: Considering the complexity of neurological assessment, was the examiner trained or specialized? Consider providing this information.

• The Pediatric Stroke Outcome Measure was designed to assess outcomes after stroke in infants and children. It would be interesting to include a justification for using this assessment tool.

RESULTS

• Lines 89/91 and 167: In the methods section, the age group is described as 18-50 years and gestational age <37 weeks. However, in table 1 the age group is from 18 to 45 years old, and the gestational age is up to 39.4 weeks. Could the authors check it out?

• Line 174: In the text it shows the Hispanic ethnicity at almost 40% (26/67), but in the table it is 34% (26/76). Could the authors also check? In addition, as Table 1 describes the study population, I suggest including the other ethnic groups.

• Line 224 - Table 5 is missing.

• Lines 239-241: The adjusted values shown in this part do not seem close to significance as stated by the authors. Consider reviewing this sentence.

GENERAL

• Consider adding legend to all tables as appropriate.

• Line 355-358: I guess this paragraph is misplaced.

• Figure 1 is distorted. Perhaps it could be enhanced to improve the visualization by the readers.

Reviewer #2: This reviewer believes that the authors have adequately addressed all comments, except the need for data availability.

According to PLOS One guidelines, authors must share the data required to replicate all study findings reported in the article.

Although, authors have claimed that study participants did not give consent for public availability, this reviewer believes that this constitutes a flaw in the informed consent questionnaire for research purposes.

Yet, according to PLOS ONE guidelines, for studies involving human research participant data or other sensitive data, authors should be able to share de-identified or anonymized data as supplementary file. There are several methodologies for doing so, and it is of up most importance. Otherwise, it is quite difficult to replicate the study findings comprised in this manuscript.

If for a greater reason data cannot be publicly shared, data sets should be made available upon request. Due to data, this reviewer suggests this information is made clear by the authors within the methods section.

Otherwise, this reviewer is satisfied with the changes made in the manuscript and would recommend it for publication.

7. PLOS authors have the option to publish the peer review history of their article (what does this mean?). If published, this will include your full peer review and any attached files.

Reviewer #1: No

Reviewer #2: **Yes: **Livia Rosa Fernandes

---

## [Author Response · Author response to Decision Letter 1]

1 Nov 2021

INTRODUCTION

• Line 54-60: The word preeclampsia is used six times in this paragraph. Consider using synonyms.

- Agree. This paragraph has been edited.

• Review the citations made in the introduction. A few sentences have no citation.

- We have edited and confirmed all appropriate citations.

METHODS

• Line 105-106: The accuracy of gestational age (GA) is important. So, ultrasound in the first trimester of gestation can represent a better tool to access GA. Did the authors have that information? This can be added as a limitation of the study. 

-This has been added as a limitation in lines 341-343. Given our Labor and Delivery population (different arenas of prenatal care) and clinical data collection methods, these data were not available.

• Line 141: Considering the complexity of neurological assessment, was the examiner trained or specialized? Consider providing this information. 

- Dr. Armstrong is a known specialized expert in neonatal and vascular neurology. This has been noted in the methods.

• The Pediatric Stroke Outcome Measure was designed to assess outcomes after stroke in infants and children. It would be interesting to include a justification for using this assessment tool. 

- Added and PSOM included in Supplemental Material.

RESULTS

• Lines 89/91 and 167: In the methods section, the age group is described as 18-50 years and gestational age <37 weeks. However, in table 1 the age group is from 18 to 45 years old, and the gestational age is up to 39.4 weeks. Could the authors check it out? 

- These numbers are correct. Per the methods study inclusion criteria were women 18-50 years old, while Table 1 presents the age range of the women actually consented into the study, with the oldest being 45 years old. As for gestational age, per the methods study inclusion criteria was gestational age <37 weeks at time of hospitalization for preeclampsia, and Table 1 provided gestational age at time of delivery. This has been clarified in the methods and Table 1.

• Line 174: In the text it shows the Hispanic ethnicity at almost 40% (26/67), but in the table it is 34% (26/76). Could the authors also check? In addition, as Table 1 describes the study population, I suggest including the other ethnic groups. 

- There were no other ethnicities in our study population. This is representative of the Rocky Mountain Region, which is mainly a white population non-Hispanic population. Table 1 and text explicitly updated regarding this.

• Line 224 - Table 5 is missing. 

-Thank you for catching this. We had removed this Table and instead presented as text per the recommends at the prior revision.

• Lines 239-241: The adjusted values shown in this part do not seem close to significance as stated by the authors. Consider reviewing this sentence. 

-Agree, we deleted the sentence pertaining to IL-6 as it does not approach significance.

GENERAL

• Consider adding legend to all tables as appropriate. 

-Thank you, we have discussed and have no additions for legends in the Tables. We have added additional legend information to Fig 1.

• Line 355-358: I guess this paragraph is misplaced. 

-Thank you, yes this was improperly inserted as has been removed.

• Figure 1 is distorted. Perhaps it could be enhanced to improve the visualization by the readers. 

-We have tried to enhance the Figure to the best of our ability and within the limitations of the software. If the reviewer clicks on the direct link in the constructed pdf for the uploaded .tiff Figure, the Figure is clear. It appears to be a glitch with the pdf conversion that we do not know how to fix on the author submission end.

---

## [Editor Report · Decision Letter 2]

3 Nov 2021

Inflammatory cytokines, placental pathology, and neurological outcomes in infants born to preterm preeclamptic mothers

PONE-D-21-14686R2

Dear Dr. Armstrong,

We’re pleased to inform you that your manuscript has been judged scientifically suitable for publication and will be formally accepted for publication once it meets all outstanding technical requirements.

Kind regards,

Khursheed Iqbal, Ph.D

Academic Editor

PLOS ONE

---

## [Editor Report · Acceptance letter]

5 Nov 2021

PONE-D-21-14686R2 

Inflammatory cytokines, placental pathology, and neurological outcomes in infants born to preterm preeclamptic mothers 

Dear Dr. Armstrong:

I'm pleased to inform you that your manuscript has been deemed suitable for publication in PLOS ONE. Congratulations! Your manuscript is now with our production department. 

Kind regards, 

on behalf of

Dr. Khursheed Iqbal 

Academic Editor

PLOS ONE